# Antipsychotic Medication Influences the Discriminative Value of Acylethanolamides as Biomarkers of Substance Use Disorder

**DOI:** 10.3390/ijms24119371

**Published:** 2023-05-27

**Authors:** Jesús Herrera-Imbroda, María Flores-López, Nerea Requena-Ocaña, Pedro Araos, Jessica Ropero, Nuria García-Marchena, Antonio Bordallo, Juan Suarez, Francisco Javier Pavón-Morón, Antonia Serrano, Fermín Mayoral, Fernando Rodríguez de Fonseca

**Affiliations:** 1Grupo de Neuropsicofarmacología, IBIMA-Plataforma BIONAND, 29590 Málaga, Spain; 2Unidad de Gestión Clínica de Salud Mental, Hospital Regional Universitario de Málaga, IBIMA-Plataforma BIONAND, Hospital Regional Universitario de Málaga, 29010 Málaga, Spain; 3Facultad de Medicina, Universidad de Málaga, Andalucía Tech, Campus de Teatinos s/n, 29071 Málaga, Spain; 4Departamento de Psicología Básica, Facultad de Psicología, Universidad de Málaga, 29071 Málaga, Spain; 5Departamento de Psicobiología y Metodología, Facultad de Psicología, Universidad Complutense de Madrid, 28223 Madrid, Spain; 6Departamento de Anatomía, Medicina Legal e Historia de la Ciencia, Facultad de Medicina, Universidad de Málaga, 29071 Málaga, Spain; 7Unidad Clínica Área del Corazón, Hospital Universitario Virgen de la Victoria, IBIMA-Plataforma BIONAND, 29010 Málaga, Spain; 8Centro de Investigación Biomédica en Red de Enfermedades Cardiovasculares (CIBERCV), Instituto de Salud Carlos III, 28029 Madrid, Spain; 9Unidad Clínica de Neurología, Hospital Regional Universitario de Málaga, IBIMA-Plataforma BIONAND, 29010 Málaga, Spain; 10Andalusian Network for Clinical and Translational Research in Neurology [NEURO-RECA], 29001 Málaga, Spain

**Keywords:** substance use disorders, biomarkers, endocannabinoids, acylethanolamides, neuroleptics, psychiatric co-morbidity

## Abstract

Plasma acylethanolamides (NAEs), including the endocannabinoid anandamide (AEA), have been proposed as circulating biomarkers of substance use disorders. However, the concentration of these lipid transmitters might be influenced by the use of drugs prescribed for either the treatment of addiction or the associated psychiatric co-morbidities such as psychosis. As an example, neuroleptics, used for attenuation of psychotic symptoms and sedation, might theoretically interfere with the monoamine-mediated production of NAEs, obstructing the interpretation of plasma NAEs as clinical biomarkers. To solve the lack of information on the impact of neuroleptics on the concentration of NAEs, we evaluated the concentrations of NAEs in a control group and compared them to those present in (a) substance use disorders (SUD) patients that are not prescribed with neuroleptics, and (b) SUD patients (both alcohol use disorder and cocaine use disorder patients) using neuroleptics. The results demonstrate that SUD patients exhibited greater concentrations of NAEs than the control population, affecting all species with the exception of stearoylethanolamide (SEA) and palmitoleoylethanolamide (POEA). Neuroleptic treatment enhanced the concentrations of NAEs, especially those of AEA, linoleoylethanolamide (LEA), and oleoylethanolamide (OEA). This effect of neuroleptic treatment was observed independently of the drug addiction that motivated the demand for treatment (either alcohol or cocaine). This study remarks the need to control the current use of psychotropic medication as a potential confounding variable when considering the use of NAEs as biomarkers in SUD.

## 1. Introduction

Drug addiction is a chronic and recurrent mental disorder characterized by compulsive drug seeking despite serious negative consequences [1]. Substance use disorders include several diagnostic categories in the Diagnostic and Statistical Manual of Mental Disorders, Fifth Edition (DSM-5) [2], in which patients must meet a series of criteria that distinguish them from the occasional use of drugs of abuse. Substance use disorder (SUD) is a major public health problem, with 0.5 million deaths attributable to illicit drug use annually and an estimated 35 million people suffering from drug use disorders around the world [3]. Multiple neural networks in the brain, including the reward system, the anti-reward/stress system, and the central immune system, are involved in the development of these disorders [4]. Because of its wide impact on physiological systems, SUD is frequently associated with multiple co-morbid disorders, including psychiatric diseases, that complicate diagnosis and hinder treatment options. Stratification of patients is thus a major challenge in SUD because the drug used and the co-morbidities present at diagnosis are insufficient to address acceptable outcomes of standard treatments [5]. The search for biomarkers for optimizing the clinical management of SUD patients led to the evaluation of multiple biochemical signatures, including the endocannabinoid system [6,7].

The endocannabinoid system (ECS) is defined as a lipid-based neuromodulatory system involved in the control of synaptic transmission, bioenergetic adjustment, and plasticity/inflammation/repair roles [8,9,10]. Altogether, these functions make the ECS an important modulator of various homeostatic functions such as eating, reproduction, social behavior, play, learning, memory, and stress responses. The ECS comprises receptors, endogenous ligands, and the machinery for its biosynthesis and degradation [8,9]. The two cannabinoid receptors identified to date are G-protein coupled receptors with seven transmembrane domains: cannabinoid receptor type 1 (CB1R) and cannabinoid receptor type 2 (CB2R). CB1R is most prominent in the central nervous system (CNS) and is associated with several psychiatric and neurological disorders, being the psychoactive target for Δ^9^—tehtrahydrocannabinol [9,10]. The main endogenous ligands of endocannabinoid receptors are anandamide or arachidonoylethanolamide (AEA) and 2-arachidonolylglycerol (2-AG), both of them derivatives of arachidonic acid [8,11]. Structurally related anandamide-like compounds are the acylethanolamides or N-acylethanolamines (NAEs). Most of them do not bind to cannabinoid receptors but can modify AEA bioavailability because they, in several cases, engage the same synthesizing and degrading enzymes [8,9]. Non-cannabinoid NAEs can exert additional homeostatic functions [12] through the interaction with other receptors such as the PPARα nuclear receptor [13], the ionotropic Vanilloid VR1 receptor [14], or the orphan receptor GPR119 [15]. Some of the most studied NAEs are oleoylethanolamide (OEA), palmitoylethanolamide (PEA), linoleoylethanolamide (LEA), and stearoylethanolamide (SEA) [8,16]. Interestingly, NAE production has been related to monoamine and glutamatergic activities. Thus, activation of dopamine D2-type receptors enhances the production of AEA to attenuate dopamine-associated behavioral activation typical of both SUD and psychosis [17]. Similarly, postsynaptic 2-AG release in glutamatergic synapsis acts as a retrograde mechanism engaged in reducing the enhanced glutamatergic response associated with SUD and psychosis [8,17,18].

It has been reported that the ECS may mediate biological responses associated with the pharmacological actions of substances of abuse, including alcohol [18] and cocaine [19]. In this sense, some of its species (AEA, OEA) have been proposed as potential biomarkers in SUDs [6,7]. This relationship has also been studied for other highly prevalent mental disorders that associate with SUD, such as depression, post-traumatic stress disorder, or psychosis disorders [20,21,22,23,24]. In general, the results of all these studies show an increased tone of the ECS in people with these mental disorders. This increased activation, especially in the early stages of the disease [25], may represent an attempt by the organism to counteract the neurophysiological changes triggered by the disease.

However, the value of monitoring plasma NAEs as biomarkers of co-morbid psychiatric disorders in SUD might be potentially hindered by psychotropic medication since the mechanism of NAE production can be mediated by monoamines, the main target of drugs used to treat major psychiatric disorders. This interaction has been identified in depressive patients treated with serotonin-selective reuptake inhibitors [20]. Consequently, it is necessary to discriminate whether this enhanced cannabinoid tone proposed in SUD might be affected by the concomitant use of psychiatric medication. In the present work, we analyzed whether there are alterations in the ECS in patients.

With antipsychotic treatment and SUD in comparison with patients with SUD that do not use this medication. Thus, the main aim of this study was to determine the plasma concentration of NAEs in a cohort of abstinent patients with lifetime SUD who were recruited from outpatient treatment programs, stratifying them by the use of atypical antipsychotic medication prescribed.

## 2. Results

### 2.1. Sociodemographic Characteristics and Plasma Concentration of NAEs of Control and SUD Populations

A sociodemographic and clinical description of the total sample (*n* = 508) is shown in Table 1. A total of 508 subjects were included according to the eligibility criteria and grouped into SUD (*n* = 333) and control (*n* = 175) groups. In the SUD group, the abstinent patients with SUD showed a mean age of 43.50 years, a mean BMI of 26.35 kg/m^2^, and 79.6% were men. A control group showed a mean age of 40.65 years, a mean body mass index (BMI) of 24.80 kg/m^2,^ and 52.8% were men. We observed significant differences between both groups in age, BMI, and sex. For this reason, we controlled these variables in other analyses. 

All NAE concentrations, except for POEA and SEA, were significantly different in both groups using non-parametric tests (Table 2). Thus, patients with SUD had significantly higher AEA, DEA, DGLEA, LEA, OEA, and PEA (*p* < 0.01) concentrations than the control subjects. All these differences were maintained as statistically significant after the Sidak’s correction test (*p* < 0.0057).

### 2.2. Characteristics of the SUD Group Based on Antipsychotic Treatment: Impact on Plasma Concentrations of NAEs

Table 3 shows a sociodemographic and clinical description of the 333 participants with SUD based on the use of atypical neuroleptics (SUD without antipsychotics and SUD with antipsychotics). In addition, Table 4 shows plasma concentrations of NAEs in the SUD and SUD + antipsychotic groups.

In relation to sociodemographic variables, in the SUD group (without antipsychotics), the patients showed a mean age of 44.11 years, a mean BMI of 26.25 kg/m^2^, and 79.9% were men. In the SUD + antipsychotics group, the patients showed a mean age of 39.05 years, a mean BMI of 27.13 kg/m^2,^ and 80% were men. In relation to psychiatric co-morbidity, we observed a higher percentage in the SUD + antipsychotics group for all categories: mood disorders, anxiety disorders, psychotic disorders, personality disorders, ADHD, and two or more psychiatric disorders. In relation to SUDs, we observed a higher percentage in the SUD + antipsychotics group for cocaine, cannabis, and two or more substances; and we observed a higher percentage in the SUD group for alcohol. In relation to the use of other psychotropic medication, we observed a higher percentage in the SUD + antipsychotics group for antidepressants and anxiolytics; and we observed a higher percentage in the SUD group for disulfiram. 

As expected, significant differences were observed between sample groups (SUD and SUD + antipsychotic treatment) with respect to age (*p* ≤ 0.05) and psychiatric co-morbidity, especially in anxiety disorders and psychotic disorders (*p* < 0.001). In co-morbid with other SUDs, the results showed significant differences in CUD, cannabis use disorders, and two or more SUDs. Thus, patients with SUD and antipsychotic treatment have more co-morbidity with substance use and psychiatric disorders. Moreover, we observed significant differences in psychotropic medication between both groups. The SUD and antipsychotic treatment group was prescribed more psychiatric medication. 

Additionally, the comparison of these groups based on the use of atypical neuroleptics and the control group revealed significant differences in NAEs using non-parametric analysis, except for POEA and SEA (Appendix A). However, we observed significant differences in AEA, DEA, DGLEA, LEA, and OEA between the SUD and SUD + antipsychotics groups when using a non-parametric test (Table 4). Thus, patients with SUD and antipsychotic treatment had significantly higher plasma concentrations of these NAE (*p* < 0.05) concentrations than SUD subjects. After the Sidak’s correction test, statistical significance was only maintained in the case of AEA (*p* < 0.0057). 

In addition, we analyzed the concentration of NAEs between groups with one-way ANCOVA. Raw data for plasma concentrations of NAEs were log10-transformed to ensure statistical assumptions of the one-way ANCOVA while controlling for age, BMI, and sex. Back-transformed values of the estimated marginal means and 95% CI of log10-transformed NAE were represented (Appendix A and Figure 1). The analysis revealed a significant main effect of the group factor (SUD and SUD + antipsychotic treatment) on AEA [F (1, 333) = 4.314, *p* = 0.039] (Figure 1A), LEA [F (1, 333) = 5.605, *p* = 0.018] (Figure 1C), and OEA [F (1, 306) = 6.382, *p* = 0.012] (Figure 1D) concentrations, which confirmed some of the previous differences (Table 4). 

### 2.3. Plasma Concentrations of NAEs Based on Type of SUD and Antipsychotic Treatment

Plasma concentrations of NAEs were examined according to the diagnosis of lifetime SUD and antipsychotic treatment. For this purpose, raw data for NAEs were log10-transformed to ensure statistical assumptions of the two-way ANCOVA, with the type of SUD and antipsychotic treatment as factors, while controlling for sex, age, and BMI. Figure 2 and Appendix A show the back-transformed values of the estimated marginal means and 95% CI of log10-transformed AEA, DEA, DGLEA, LEA, OEA, PEA, POEA, and SEA based on a diagnosis of type SUD and antipsychotic treatment.

We defined four groups according to the diagnosis of lifetime SUD and atypical neuroleptics use: AUD (without antipsychotics) group, CUD (without antipsychotics) group, AUD + antipsychotics group, and CUD + antipsychotics group. LEA and POEA concentrations were higher in the AUD group than in the CUD group; and AEA, DEA, DGLEA, DHEA, OEA, PEA, and SEA concentrations were higher in the CUD group than in the AUD group. On the other hand, LEA, OEA, PEA, and POEA concentrations were higher in the AUD + antipsychotics group than in the CUD + antipsychotics group; and AEA, DEA, DGLEA, DHEA, and SEA concentrations were higher in the CUD + antipsychotics group than in the AUD + antipsychotics group. Finally, we found higher concentrations of all NAEs both in the AUD and CUD groups when the use of antipsychotics was added, except for OEA and PEA in the case of the CUD group.

Specifically, the two-way ANCOVA revealed a significant main effect of the type of SUD on AEA [F (2, 333) = 4.281, *p* = 0.015] (Figure 2A) and SEA [F (2, 232) = 3.855, *p* = 0.023] (Appendix A) concentrations. Regarding the treatment with antipsychotics, we observed a significant main effect of this factor on AEA [F (1, 333) = 4.459, *p* = 0.035] (Figure 2A), LEA [F (1, 333) = 7.841, *p* = 0.005] (Figure 2C), and OEA [F (1, 306) = 4.259, *p* = 0.040] (Figure 2D). After the Sidak’s correction test, statistical significance was only maintained in the case of LEA (*p* < 0.0057). However, there were no significant interactions between the type of SUD and the treatment with antipsychotics on the plasma concentrations of NAEs in the remaining species.

### 2.4. Plasma Concentration of NAEs as Predictors of Antipsychotic Treatment

A first logistic regression model for the discrimination of patients with SUD from healthy control subjects was constructed using all NAEs (log10-transformed concentrations), age, BMI, and sex (Appendix A). The ROC analysis revealed an excellent discriminative power for patients with SUD [AUC = 0.900 (95% CI = 0.845–0.954), *p* < 0.001)] (Figure 3A).

In a similar way, another logistic regression model for distinguishing patients with SUD from patients with SUD without antipsychotic treatment and antipsychotic treatment was performed using all NAEs (log10-transformed concentrations), age, BMI, and sex (Appendix A). In this case, the ROC analysis indicated a significant discriminative power of the model [AUC = 0.807 (95% CI = 0.706-0.908), *p* < 0.001] (Figure 3B) and representative cutoff values showed high sensitivity and specificity [for example, 0.1367 (83.3% sensitivity and 69.7% specificity) and 0.1399 (83.3% sensitivity and 70.6% specificity) for SUD].

## 3. Discussion

Two major findings emerge from the present study: first, confirmation of the value of monitoring NAEs in plasma in SUD as potential biomarkers; and second, the influence of psychiatric medications (in this case, atypical neuroleptics) on circulating blood levels of these lipid mediators. Regarding the value of NAEs as biomarkers for SUD, the present study confirms, with a much larger sample, previous reports identifying the existence of an elevated endogenous cannabinoid tone in SUDs (mainly AUD and/or CUD) patients demanding treatment [6,7]. However, we did not evaluate acylglycerols in the present study. As reported previously, we have replicated the increases in AEA, the more important NAE activating cannabinoid receptors [8]; OEA, which is capable of reducing drug seeking-behavior [26,27] and alcohol-induced neuroinflammation [28]; as well of that as LEA, an anti-inflammatory NAE capable of boosting AEA/OEA concentrations by interfering with fatty acid amidohydrolase-mediated degradation of NAEs [29]. In addition, other less studied NAEs, such as DHEA, capable of promoting neuronal differentiation of the hippocampus [30], or DGLEA, which displays an affinity for cannabinoid receptors [31] were found to be also elevated in abstinent SUD patients, suggesting that the NAE signaling system participants in the adaptive responses to the pro-inflammatory actions of abused drugs. This effect is thought to occur through several mechanisms that involve CB1R, CB2R [8,9], PPARα, and PPARγ receptors [32,33], or through the conversion of NAEs into endoperoxides with anti-inflammatory properties [34]. Interestingly, some of these mechanisms are involved in the counter-regulatory mechanisms whose disruption is associated with psychiatric co-morbidity. Thus, OEA-induced attenuation of neuroinflammation can reduce depressive-like behaviors through the attenuation of NFkB-mediated inflammatory responses [35,36], whereas AEA is capable of attenuating the characteristic disorganized behaviors associated with the administration of psychostimulant or dopamine receptor agonists [17,26]. Interestingly, there is solid evidence demonstrating the relationship between plasma levels of these NAEs, especially OEA, and alcohol-induced neuroinflammatory damage [37]. Interestingly, the type of SUD (AUD or CUD) only affected AEA and SEA concentrations, being the increment in plasma NAEs equivalent across stratified SUD patients.

The meaning of increase in circulating NAEs observed in SUD patients is still a matter of active research. It might represent an adaptive response to (a) the drug consumption itself, thus being a state biomarker; (b) the underlying psychiatric disorders (i.e., SUD + psychosis, thus becoming a trait biomarker); or (c) the pharmacological effect of the psychiatric medication. Whether NAE concentration reflects either a trait of the SUD patients or a particular state (i.e., a biomarker of a certain stage of the SUD disease) is still a matter of debate. In any case, the answer to this question does not affect the intrinsic value of the association of these NAE concentrations to SUD diagnosis. However, an important question remains concerning the impact that psychiatric medication may have on NAE concentrations. This is relevant because a modulatory role of the medication on NAE concentrations might influence its value as a relevant clinical biomarker on SUD patients. The interaction between the ECS/NAEs and psychiatric medication is complex. On the one hand, the administration of psychiatric medication might induce an enhancement of the ECS, which indeed could contribute to the fight against the disease until resolution in some cases if therapeutic success is eventually achieved. Interestingly, a recent report identified how these NAEs were induced by antidepressants, helping them to improve somatic symptomatology [20]. Additionally, AEA has been found to be increased in the CSF, but not serum, of patients with acute schizophrenia, and its levels are associated with behavioral symptomatology and prognosis, supporting this protective role [25,38,39,40]. However, the impact of antipsychotics, especially atypical neuroleptics, on plasma NAEs is not well understood and needs further research. As an example of the complexity, in the study of Leweke and colleagues on schizophrenia patients, while patients with psychotic symptoms displayed high AEA levels in the CSF, with respect to the control population, the use of typical neuroleptics (dopamine D-2 receptor-preferring drugs) decreased them, while the treatment with atypical neuroleptics (that also display an affinity for other targets such as serotonin 5HT_2A_ receptors) did not lower the enhanced AEA levels. This profile is different from that observed in our patients, where we monitored only plasma concentrations of NAEs. Thus, the pharmacological profile of the drug, dose, length of treatment, stage of the disease, co-morbid disorders associated, and anatomical compartment (i.e., brain versus plasma) may have a relevant influence in the final direction of NAE production observed.

Nonetheless, these previous reports suggest that the interference of psychiatric medication might hinder the use of NAEs/endocannabinoids as specific biomarkers, especially in case of drug-induced changes in their circulating levels that might variate in the opposite direction to that reported to be associated by SUD, in this case, an enhancement of NAE tone. Additionally, this is a relevant problem because psychiatric medication is really frequent in SUD patients [6,7]. In the substance abuse user population of the present, we found more prescriptions of antipsychotic drugs, which can be explained in part by the high co-morbidity between SUDs and psychosis [41,42]. However, it is also known that off-label medications are often used in SUD, while adoption rates of drugs specifically approved for this indication remain low. Moreover, among these off-label medications, antipsychotic drugs precisely occupy a prominent place [43]. This could perhaps be due to the anti-impulsive and anxiolytic effects that some of the antipsychotic drugs may have in this population, even if they have not developed psychosis disorders. Thus, it is important to control how these drugs affect the circulating levels of NAEs. Our results clearly demonstrate that this type of medication further increases the concentration of circulating levels of NAEs, specially AEA, OEA, and LEA, with a clear antipsychotic/anti-inflammatory profile, supporting the above-described hypothesis of a protective role for these lipid mediators [25]. We must stress that there are still numerous clinical trials exploring the utility of neuroleptics in co-morbid SUD/psychosis patients [44] and that new ECS-acting drugs (i.e., inhibitors of endocannabinoid degradation) are also being considered for the treatment of this association [45]. The fact that the increase in circulating NAE concentrations induced by neuroleptics allows the differentiation of these patients from those SUD patients is sufficiently relevant to consider future evaluations of NAEs as biomarkers of response to neuroleptic treatments, with the above-discussed restrictions.

In any case, the present study has several important limitations to highlight. First, this is a transversal evaluation of a SUD population, and we have no indication of the evolution of these patients regarding both NAE biomarkers and clinical response to treatment. Second, the number of psychotics is low (despite psychosis being a much more frequent diagnosis in SUD patients when compared to the general healthy population). Third, this low number precluded both a detailed analysis of the specific neuroleptic impact and a gender approach that should be addressed in future studies. Finally, analysis of neuroleptic-free psychosis in SUD patients (first episodes) is needed to clarify the impact of the disease on the circulating levels of NAEs.

## 4. Materials and Methods

### 4.1. Participants and Recruitment

The present study included 508 Caucasian volunteers divided into two groups: 175 healthy control subjects and 333 abstinent SUD patients in outpatient treatments. Patients were recruited at the Centro Provincial de Drogodependencias (Málaga, Spain) and the outpatient alcohol program at the Hospital Universitario 12 de Octubre (Madrid, Spain). Control participants were included from databases of healthy subjects willing to participate in medical research projects from multidisciplinary staff working at the Hospital Regional Universitario de Malaga (Málaga, Spain), Hospital Universitario 12 de Octubre, and Universidad Complutense de Madrid (Madrid, Spain).

To be eligible for the present study, participants had to meet the following inclusion criteria: ≥18 years to 65 years of age and abstinence from alcohol and/or cocaine in the evaluation moment. The exclusion criteria included: a personal history of long-term inflammatory diseases or cancer, cognitive or language limitations, pregnant or breast-feeding women, and infectious diseases. Prescription of antipsychotic medication was obtained from clinical records, and only patients with atypical neuroleptic prescription were included (including olanzapine, clozapine, quetiapine, aripiprazole, paliperidone, and risperidone). Regarding the control group, participants with psychiatric disorders or psychotropic drug consumption were also excluded.

### 4.2. Ethics Statements

Written informed consent was obtained from each participant after a complete description of the study. All the participants had the opportunity to discuss any questions or issues. The study and protocols for recruitment were approved by the Ethics Committee of the Hospital Regional Universitario de Málaga (PND2018I033, approved 25 October 2018) in accordance with the Ethical Principles for Medical Research Involving Human Subjects adopted in the Declaration of Helsinki by the World Medical Association (64th WMA General Assembly, Fortaleza, Brazil, October 2013), the Recommendation No. R (97) 5 of the Committee of Ministers to Member States on the protection of medical data (1997), and the Regulation (EU) 2016/679 of the European Parliament and the Council 27 April 2016 on the protection of natural persons concerning the processing of personal data and on the free movement of such data (General Data Protection Regulation). All collected data were given code numbers in order to maintain privacy and confidentiality.

### 4.3. Clinical Assessments

SUDs and other psychiatric disorders were diagnosed according to the DSM-IV-TR criteria (APA, 2000) using the Spanish version of the Psychiatric Research Interview for Substance and Mental Disorders (PRISM). PRISM is a semi-structured interview with good psychometric properties in the evaluation of SUDs and the main psychiatric co-morbid disorders related to the substance use population [46].

### 4.4. Collection of Plasma Samples

Blood samples were obtained in the morning after fasting for 8–12 h (prior to the psychiatric interviews). Venous blood was extracted into 10 mL K_2_ EDTA tubes (BD, Franklin Lakes, NJ, USA) and immediately processed to obtain plasma. Blood samples were centrifuged at 2200× *g* for 15 min (4 °C) and individually assayed to detect infectious diseases by four commercial-rapid tests for HIV, hepatitis B, and hepatitis C (Master Labor SL, Madrid, Spain) and SARS-CoV-2 (Bio-Connect, Huissen, The Netherlands). Finally, plasma samples were individually characterized, registered, and stored at −80 °C until further analyses.

### 4.5. Quantification of Endocannabinoids in Plasma

The analysis of NAEs in plasma was performed by a validated method previously described (Pastor et al., 2014). The following NAEs were quantified: palmitoylethanolamide (PEA), stearoylethanolamide (SEA), oleoylethanolamide (OEA), palmitoleoylethanolamide (POEA), arachidonoyl-ethanolamide (AEA), linoleoylethanolamide (LEA), docosahexaenoylethanolamide (DHEA), dihomo-γ-linolenoylethanolamide (DGLEA), and docosatetraenoyl-ethanolamide (DEA).

Briefly, aliquots of 0.5 mL of human plasma were transferred to 12-mL glass tubes, spiked with deuterated internal standards, diluted with 0.1 M ammonium acetate buffer (pH 4.0), and extracted with a tert-butyl methyl ether. The dry organic extracts were reconstituted in 100 μL of a mixture water:acetonitrile (10:90, *v*/*v*) with 0.1 percent formic acid (*v*/*v*) and transferred to HPLC vials. Twenty microliters were injected into the LC/MS-MS system. An Agilent 6410 triple quadrupole (Agilent Technologies, Wilmington, DE, USA) equipped with a 1200 series binary pump, a column oven, and a cooled auto-sampler (4 °C) was used. Chromatographic separation was carried out with an ACQUITY UPLC C18-CSH column (3.1 × 100 mm, 1.8-μm particle size) (Waters, Yvelines Cedex, France) maintained at 40 °C with a mobile phase flow rate of 0.4 mL/minute. The composition of the mobile phase was: A: 0.1 percent (*v*/*v*) formic acid in water; B: 0.1 percent (*v*/*v*) formic acid in acetonitrile. Quantification was performed by isotope dilution. Deuterated internal standards were obtained from Cayman (Cayman Chemical, Ann Arbor, MI, USA), and solvents were from Merck (Merck, Darmstadt, Germany).

### 4.6. Statistical Analysis

Date in Table 1, Table 2, Table 3 and Table 4 were expressed as the number and percentage of the subject (*n* (%)), mean and standard deviation (SD), or median and interquartile range (IQR). Statistical differences in categorical variables were evaluated with the chi-square test or Fisher’s exact test, whereas differences in continuous variables were evaluated with the Student’s *t*-test for a normal distribution or the Mann–Whitney U test for a non-normal distribution.

Analysis of covariance (ANCOVA) (Figure 1 and Figure 2) was used to evaluate the main effects and interaction of primary independent variables (group/subgroup factor) (i.e., control and SUD; SUD and SUD + Antipsychotic treatment) on NAE concentrations while adjusting for age, BMI and sex as covariates. Raw data for NAE concentrations were log10-transformed because their distribution was positively skewed to ensure statistical assumptions of the ANCOVA. Post hoc comparisons for multiple comparisons were performed using Sidak’s correction test. The estimated marginal means and 95% confidence interval (95% CI) of log10-transformed NAE concentrations were back-transformed in the figures. Receiver operating characteristics (ROC) analyses were performed to evaluate the discriminative power of binary logistic regression models through the area under the curve (AUC). In addition, the resulting probability data from these models were compared between groups/subgroups using the Student’s t-test or Mann–Whitney U test.

The GraphPad Prism version 5.04 (GraphPad Software, San Diego, CA, USA) and IBM SPSS Statistics version 22 (IBM, Armonk, NY, USA) were used for the statistical studies. A *p*-value of less than 0.05 was considered statistically significant. In the case of post hoc comparisons for multiple comparisons using Sidak’s correction test, a *p*-value of less than 0.0057 was considered statistically significant (1 − [1 − 0.05]^1/9^).

## 5. Conclusions

Our study confirmed three important findings: first, the circulating levels of various NAEs in the plasma of SUD patients are elevated when compared with those of healthy controls; second, there is a clear influence of neuroleptic treatment in the circulating levels of three relevant NAEs: AEA, OEA, and LEA; and finally, three, the type of SUD (alcohol or cocaine) also influences the impact of neuroleptics on plasma circulating levels of NAES. Overall, these differences are sufficiently relevant to generate a predictive model that discriminates neuroleptic-using SUD patients from SUD patients not using this type of medication, eventually serving as a biomarker of adherence to the neuroleptic treatment in SUD patients.

## Figures and Tables

**Figure 1 ijms-24-09371-f001:**
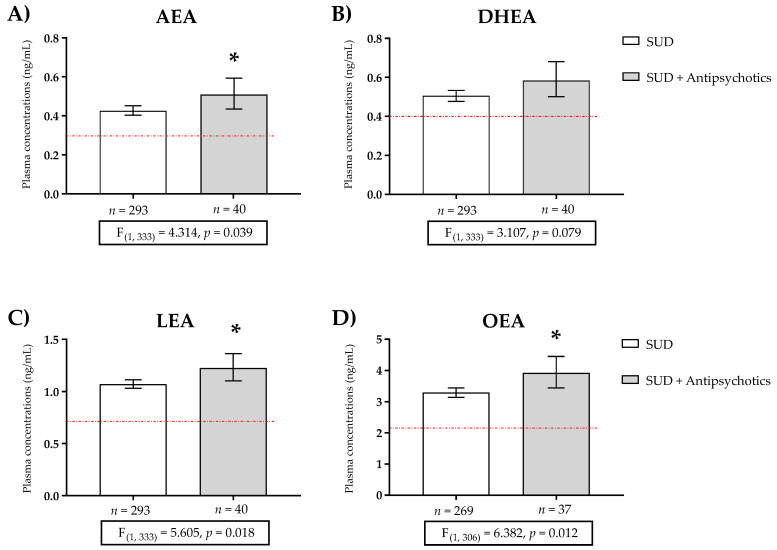
Plasma concentrations of (**A**) anandamide (AEA), (**B**) docosahexaenoylethanolamide (DHEA), (**C**) linoleoylethanolamide (LEA), and (**D**) oleoylethanolamide (OEA) in patients with substance use disorder (SUD) not using antipsychotics, and patients of SUD using antipsychotics. The red line represents the mean plasma concentration of the control group. Data were analyzed by one-way analysis of covariance (ANCOVA). Bars are estimated as marginal means and 95% confidence intervals (95% CI). * *p* < 0.05 versus SUD group.

**Figure 2 ijms-24-09371-f002:**
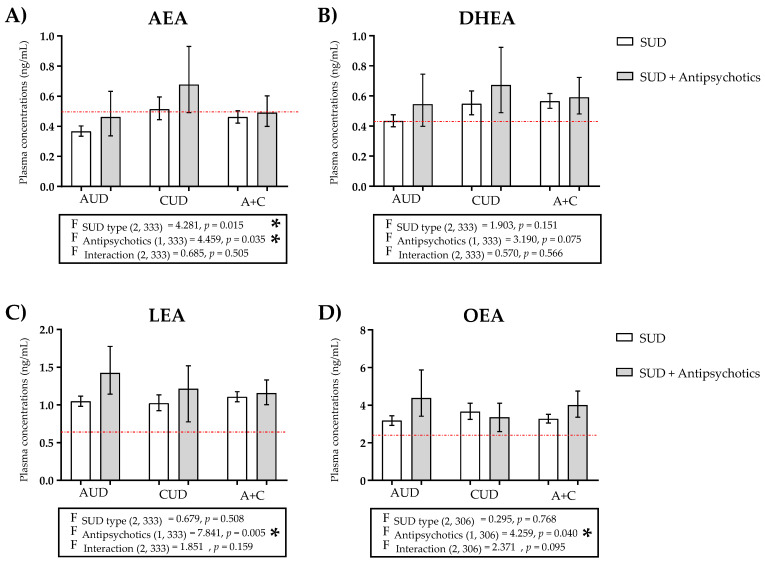
Plasma concentrations of (**A**) AEA, (**B**) DHEA), (**C**) LEA, and (**D**) OEA in patients with SUDs not using antipsychotics, and patients of SUD using antipsychotics classified on the basis of AUD, CUD, or AUD + CUD diagnosis. The red line represents the mean plasma concentration of the control group. Data were analyzed by two-way ANCOVA using the SUD type and antipsychotics as factors. Bars are estimated as marginal means and 95% CI. * *p* < 0.05 for factors (F).

**Figure 3 ijms-24-09371-f003:**
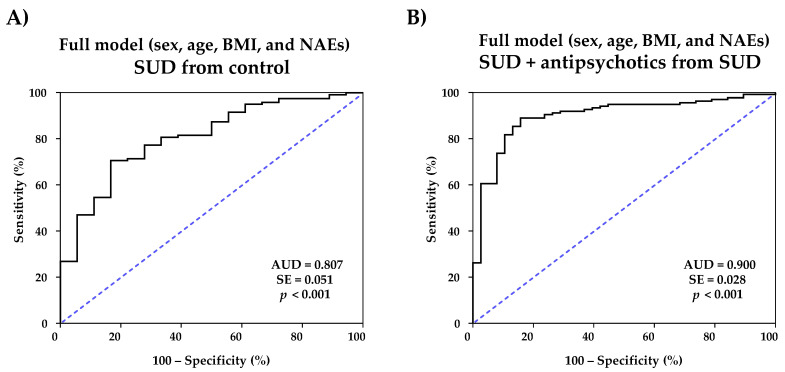
ROC analyses of predictive multivariate full models using sex, age, BMI, and plasma concentrations of NAEs for (**A**) distinguishing SUD patients from control population; and (**B**) distinguishing SUD + antipsychotic patients from SUD patients.

**Table 1 ijms-24-09371-t001:** Baseline sociodemographic characteristics of control and SUD populations.

Variable		Control (*n* = 175)	SUD (*n* = 333)	*p* Value
**Sex** [*n* (%)]	Women	80 (44.4)	67 (20.1)	**0.000 ^1^**
	Men	95 (52.8)	266 (79.6)	
**Age**	mean ± SD	40.65 ± 12.40	43.50 ± 11.16	**0.005 ^2^**
	median (IQR)	40.0 (31.5–47.0)	43.0 (34.4–51.0)	
**BMI**	mean ± SD	24.80 ± 3.67	26.35 ± 4.85	**0.001 ^2^**
	median (IQR)	24.8 (22.31–27.0)	25.56 (22.99–29.05)	
**Marital status** [*n* (%)]	Single	82 (45.6)	116 (34.8)	**0.000 ^1^**
	Married/cohabiting	59 (32.8)	131 (39.3)	
	Divorced/separated	13 (7.2)	80 (24.0)	
	Widowed	2 (1.1)	6 (1.8)	
**Education** [*n* (%)]	≤ Primary	7 (3.9)	116 (34.8)	**0.000 ^1^**
	Secondary	55 (30.6)	170 (51.1)	
	University	102 (56.6)	47 (14.1)	
**Work status** [*n* (%)]	Employed	136 (75.5)	101 (30.3)	**0.000 ^1^**
	Unemployed	23 (12.8)	186 (55.8)	
	Sick leave	2 (1.1)	37 (11.1)	
	Housework	2 (1.1)	9 (2.7)	
**Psychiatric co-morbidity** [*n* (%)]	Mood Disorders	-	143 (42.9)	-
	Anxiety Disorders	-	95 (28.5)	-
	Psychotic Disorders	-	36 (10.8)	-
	Personality Disorders	-	85 (25.5)	-
	ADHD	-	33 (9.9)	-
	>2 psychiatric disorders	-	232 (69.7)	-
**SUDs** [*n* (%)]	Alcohol	-	281 (84.4)	-
	Cocaine	-	201 (60.4)	-
	Cannabis	-	78 (23.4)	-
	>2 substances	-	178 (53.5)	-
**Psychotropic medication** [*n* (%)]	Antidepressants	-	132 (39.6)	-
	Anxiolytics	-	155 (46.5)	-
	Antipsychotics	-	40 (12)	-
	Disulfiram	-	133 (39.9)	-
**SUD duration years** [median (IQR)]	AUD	-	10 (4–17)	-
	CUD	-	5 (2–12)	-
**Days of abstinence** [median (IQR)]	AUD	-	69 (2–210)	-
	CUD	-	25 (0.75–120)	-

(^1^) *p*-value from chi-square test; (^2^) *p*-value from Mann–Whitney U test. *p*-value in bold indicates a statistically significant difference. Abbreviations: ADHD = attention deficit hyperactivity disorder; BMI = body mass index; IQR = interquartile range; SD = standard deviation; SUD = substance use disorder; AUD = alcohol use disorder; CUD = cocaine use disorder.

**Table 2 ijms-24-09371-t002:** Plasma concentrations of NAEs in the control and SUD groups.

NAEs	Control (*n* = 180)	SUD (*n* = 333)	U-Statistic	*p* Value
**AEA** median (IQR)	0.33 (0.21–048)	0.44 (0.30–0.64)	20,219.50	**0.000**
**DEA** median (IQR)	0.08 (0.05–0.14)	0.13 (0.09–0.18)	8207.50	**0.000**
**DGLEA** median (IQR)	0.07 (0.04–0.09)	0.08 (0.06–0.12)	20,123.00	**0.000**
**DHEA** median (IQR)	0.42 (0.27–0.61)	0.52 (0.37–0.72)	20,054.50	**0.000**
**LEA** median (IQR)	0.70 (0.51–0.93)	1.08 (0.86–1.38)	11,351.00	**0.000**
**OEA** median (IQR)	1.88 (1.33–3.15)	3.37 (2.52–4.51)	11,951.50	**0.000**
**PEA** median (IQR)	1.59 (1.27–2.45)	3.37 (2.25–5.72)	9070.00	**0.000**
**POEA** median (IQR)	0.29 (0.16–0.39)	0.31 (0.19–0.43)	4752.00	0.276
**SEA** median (IQR)	4.30 (0.90–5.90)	2.06 (1.30–4.20)	18,817.50	0.775

*p*-value from Mann–Whitney U test. *p*-value in bold indicates a statistically significant difference. Abbreviations: NAEs = acylethanolamides; AEA = arachidonoylethanolamide; DEA = docosatetraenoylethanolamide; DGLEA = dihomo-γ-linolenylethanolamide; DHEA = docosahexaenoylethanolamide; LEA = linoleoylethanolamide; OEA = oleoylethanolamide; PEA = palmitoylethanolamide; POEA = palmitoleoylethanolamide; SEA = stearoylethanolamide.

**Table 3 ijms-24-09371-t003:** Clinical characteristics of the SUD (without antipsychotics) and SUD + antipsychotics groups.

Variable		SUD (*n* = 293)	SUD + Antipsychotics (*n* = 40)	*p* Value
**Sex** [*n* (%)]	Men	234 (79.9)	32 (80.0)	0.984 ^(1)^
	Women	59 (20.1)	8 (20.0)	
**Age** (mean ± SD)		44.11 ± 11.3	39.05 ± 9.05	**0.002 ^(2)^**
**BMI** (mean ± SD)		26.25 ± 4.83	27.13 ± 5.04	0.303 ^(2)^
**Psychiatric co-morbidity** [*n* (%)]	Mood Disorders	121 (41.3)	22 (55.0)	0.101 ^(1)^
	Anxiety Disorders	78 (26.6)	17 (42.5)	**0.037 ^(1)^**
	Psychotic Disorders	28 (9.6)	8 (20.0)	**0.046 ^(1)^**
	Personality Disorders	70 (23.9)	15 (37.5)	0.064 ^(1)^
	ADHD	27 (9.2)	6 (15.0)	0.257 ^(1)^
	>2 psychiatric disorders	197 (67.2)	35 (87.5)	**0.009 ^(1)^**
**SUDs** [*n* (%)]	Alcohol	250 (85.3)	31 (77.5)	0.201 ^(1)^
	Cocaine	170 (58.0)	31 (77.5)	**0.018 ^(1)^**
	Cannabis	59 (20.1)	19 (47.5)	**0.000 ^(1)^**
	>2 substances	149 (50.9)	29 (72.5)	**0.010 ^(1)^**
**Psychotropic medication** [*n* (%)]	Antidepressants	110 (34.5)	22 (55.0)	**0.034 ^(1)^**
	Anxiolytics	128 (43.7)	27 (67.5)	**0.018 ^(1)^**
	Antipsychotics	-	40 (100.0)	-
	Disulfiram	120 (41.0)	13 (32.5)	0.207 ^(1)^
**SUD duration years** [median (IQR)]	AUD	10 (4–17.5)	10 (1.5–16)	0.519 ^(3)^
	CUD	5 (1.5–11)	7 (3–12.25)	0.273 ^(3)^
**Days of abstinence** [median (IQR)]	AUD	90 (2–210)	52.5 (0–217.5)	0.395 ^(3)^
	CUD	23 (0–106)	30 (11–150)	0.310 ^(3)^

(^1^) *p*-value from chi-square test; (^2^) *p*-value from Student’s *t* test. (^3^) *p*-value from Mann–Whitney U test. *p*-value in bold indicates a statistically significant difference.

**Table 4 ijms-24-09371-t004:** Plasma concentrations of NAEs in the SUD and SUD + antipsychotic groups.

NAEs	SUD (*n* = 293)	SUD + Antipsychotic (*n* = 40)	U-Statistic	*p* Value
**AEA** median (IQR)	0.43 (0.30-.061)	0.58 (0.42–0.80)	4119.0	**0.002**
**DEA** median (IQR)	0.12 (0.09–0.17)	0.16 (0.13–0.22)	2122.0	**0.011**
**DGLEA** median (IQR)	0.08 (0.06–0.12)	0.07 (0.11–0.14)	4291.5	**0.006**
**DHEA** median (IQR)	0.52 (0.36–0.70)	0.55 (0.42–0.78)	4816.5	0.068
**LEA** median (IQR)	1.07 (0.85–1.37)	1.31 (0.99–1.61)	4291.0	**0.006**
**OEA** median (IQR)	3.32 (2.47–4.47)	3.85 (2.91–5.49)	3811.0	**0.021**
**PEA** median (IQR)	3.37 (2.26–5.65)	3.38 (2.21–6.35)	3031.5	0.633
**POEA** median (IQR)	0.30 (0.19–0.42)	0.39 (0.22–0.53)	1860.5	0.105
**SEA** median (IQR)	1.90 (1.24–4.18)	3.01 (1.6–4.36)	2624.0	0.102

*p*-value from Mann–Whitney U test. *p*-value in bold indicates a statistically significant difference.

## Data Availability

Not applicable.

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
