# Peer review of "Antipsychotic Medication Influences the Discriminative Value of Acylethanolamides as Biomarkers of Substance Use Disorder"

_ijms, 2023, doi:10.3390/ijms24119371_

Round 1

Reviewer 1 Report

In this study, the concentrations of NAEs were examined in a control group and compared with those of individuals with SUD not taking neuroleptics and with those taking neuroleptics. People with SUD had higher NAE concentrations than the control group, notably of all NAE species except for stearoylethanolamide (SEA) and palmitoleoylethanolamide (POEA). Neuroleptic treatment increased the concentrations of NAEs, especially AEA, linoleoylethanolamide (LEA), and OEA, independently of the drug addiction that motivated the demand for treatment. Therefore, the authors concluded that NAEs might be useful as biomarkers in SUD but that there is a need to control the use of psychotropic medication as a potential confounding variable.

Overall, it is an interesting study. As the endocannabinoid system has been identified as a potential target for the treatment of SUD, this study is timely and provides complementary information to the current state of knowledge.

However, the current treatment trials are not mentioned in the introduction or the discussion, although these studies support the importance of the research described in the manuscript. Therefore, I suggest adding some details on this aspect.

Furthermore, the results and conclusion need to be presented more clearly, as otherwise, the value of this scientific study will not be recognized, which would be a pity.

Please find my more detailed comments below:

Introduction

The introduction is not easy to read, contains some typos, and requires careful revision.

For example:

Line 66: “transmissionl” should read “transmission”

Line 67: This sentence should probably read: Altogether, these functions make the ECS an important modulator of various homeostatic functions such as eating, reproduction, social behavior, play, learning, memory, and stress responses.

Line 70: the “machinnery fr its biosynthesis” should read “the machinery for its biosynthesis”

Line 70-75: I assume the authors intended to write something like: “The two cannabinoid receptors identified to date are G-protein coupled receptors with 7 transmembrane domains: cannabinoid receptor type 1 [CB1R] and cannabinoid receptor type 2 [CB2R]. CB1R is most prominent in the central nervous system [CNS] and is associated with several psychiatric and neurological disorders.”

Line 76: Both anandamide and 2-AG are arachidonic acid derivatives.

Line 86: Since dopamine plays a role in SUD, the synthesis of AEA due to dopaminergic receptor activation needs to be explained better, as it is an indirect mechanism. Furthermore, glutamate also plays a role in SUD, and endocannabinoids appear to modulate glutamatergic neurotransmission. Therefore, the authors should also consider adding some information about this.

Line 92 to 94: I assume the authors intended to say something like: In general, the results of all these studies show an increased tone of the ECS in people with these mental disorders. This increased activation, especially in the early stages of the disease [REFs], may represent an attempt by the organism to counteract the neurophysiological changes triggered by the disease. The latter sentence is a strong statement. This has been shown for psychosis, e.g., “Koethe D, Giuffrida A, Schreiber D, et al. Anandamide elevation in cerebrospinal fluid in initial prodromal states of psychosis. The British Journal of Psychiatry 2009;194(4):371-2. doi:10.1192/bjp.bp.108.053843, but I am not sure if there is already enough data available supporting that this is also the case for depression and anxiety.

Furthermore, references [20-25] refer to changes in depression and psychosis but not anxiety.  

Results:

The presentation of the sociodemographic data is confusing. At first, I thought there was no information on the type of SUD as this information is only included in table 3 but not in table 1. Combining tables 1 and 3 would facilitate understanding the cohorts' characteristics. Alternatively, information on psychiatric comorbidity, substance use, and psychotropic medication for the entire SUD cohort could be added to table 1.

How long was the phase of abstinence? And how long did the included participants abuse substances before they were included in the study?

Lines 133-136: It would be easier to follow if you would immediately describe what type of changes have been observed instead of stating the results very generally and then adding a sentence like: “Thus, patients with SUD and antipsychotic treatment have more comorbidity substance use and psychiatric disorders.” This is only one example. I have recognized several times that it is possible to write a bit more to the point to facilitate the understanding of your highly interesting results.

Line 151/152: What does “which confirmed some [of] the previous differences” mean? Can you please add a more detailed description of the results? I need to check the figure to understand that the AEA, OEA, and LEA levels are higher in people with SUD + antipsychotics than in people with SUD. Please also report the post hoc Sidak results.

Line 177/178: How do the significant differences look like? Again, I need to check the figure. Please also report the post hoc Sidak results.

Discussion:

Comparable to the introduction, the discussion is not easy to read, contains some typos, and requires careful revision.

For example, the first sentence should probably read: Two major findings emerge from the present study: first, confirmation of the value of monitoring NAEs in plasma in SUD as potential biomarkers, and second, the influence of psychiatric medications (in this case, atypical neuroleptics) on circulating blood levels of these lipid mediators.

In the discussion and the abstract, it should be highlighted that the blood of abstinent individuals with CUD has been evaluated. The endocannabinoid system and, most likely, the lipidome as well are highly dynamic systems. Therefore, the disease phase will be important for interpreting the data.

Line 245: I suggest adding the paper mentioned above here as well (Koethe et al. 2009). Furthermore, protective appears to be a better choice of words than defensive. Reference 39 reported that people with schizophrenia treated with atypical antipsychotics had CSF AEA levels comparable to untreated individuals and that both patient groups showed increased AEA levels compared to healthy volunteers. In contrast, the CSF AEA levels of individuals treated with typical antipsychotics did not differ from healthy volunteers. This is in contrast to the current findings and should be discussed.

Line 264/265: This is a very strong statement. Much more research is needed to be able to draw this conclusion. At least CSF AEA levels seem to be not affected by atypical antipsychotics.

Line 271: The disease state may play an important role. It also needs to be clarified if NAE changes are state or trait markers. Different baseline NAE levels may contribute to the development of SUD.

Line 276: To assess if NAEs are valuable biomarkers in SUD, studies are needed to evaluate NAEs in CSF, as it is still being determined if peripheral NAE levels also reflect changes in the brain.

Materials and Methods:

Statistics – It appears that the p-values have not been corrected for repeated testing, which should be done to compare plasma concentrations and the multiple ANCOVAs.

Conclusion:

The conclusion is written very vaguely. I have no idea which changes were observed by reading the conclusion only.

Author Response

REFEREE 1

Comments and Suggestions for Authors

1.COMMENT: In this study, the concentrations of NAEs were examined in a control group and compared with those of individuals with SUD not taking neuroleptics and with those taking neuroleptics. People with SUD had higher NAE concentrations than the control group, notably of all NAE species except for stearoylethanolamide (SEA) and palmitoleoylethanolamide (POEA). Neuroleptic treatment increased the concentrations of NAEs, especially AEA, linoleoylethanolamide (LEA), and OEA, independently of the drug addiction that motivated the demand for treatment. Therefore, the authors concluded that NAEs might be useful as biomarkers in SUD but that there is a need to control the use of psychotropic medication as a potential confounding variable.

 Overall, it is an interesting study. As the endocannabinoid system has been identified as a potential target for the treatment of SUD, this study is timely and provides complementary information to the current state of knowledge.

However, the current treatment trials are not mentioned in the introduction or the discussion, although these studies support the importance of the research described in the manuscript. Therefore, I suggest adding some details on this aspect.

Furthermore, the results and conclusion need to be presented more clearly, as otherwise, the value of this scientific study will not be recognized, which would be a pity.

 Please find my more detailed comments below:

 Answer to Referee

We thank the referee for the comments, especially on the importance of considering ongoing clinical trials where these biomarkers might have importance. We have cited in the discussion section some examples such as those related with the use of neuroleptics on comorbid schizophrenia / substance use disorder (Studies NCT03526354, NCT00156715 or NCT00130923 )

https://clinicaltrials.gov/ct2/results?cond=Substance+Use+Disorders&term=neuroleptic&cntry=&state=&city=&dist= or the use of endocannabinoid-related drugs, including FAAH inhibitors, such as Efficacy, Safety and Tolerability of the Fatty Acid Amide Hydrolase (FAAHInhibitor PF-04457845 in Adults with DSM-5 Current Cannabis Use Disorder (CUD) (https://clinicaltrials.gov/ct2/show/NCT03386487?term=FAAH+inhibitor&cond=substance+use+disorder&draw=2&rank=1)

COMMENTS TO INTRODUCTION

The introduction is not easy to read, contains some typos, and requires careful revision.

For example:

Line 66: “transmissionl” should read “transmission”

Line 67: This sentence should probably read: Altogether, these functions make the ECS an important modulator of various homeostatic functions such as eating, reproduction, social behavior, play, learning, memory, and stress responses.

Line 70: the “machinnery fr its biosynthesis” should read “the machinery for its biosynthesis”

Line 70-75: I assume the authors intended to write something like: “The two cannabinoid receptors identified to date are G-protein coupled receptors with 7 transmembrane domains: cannabinoid receptor type 1 [CB1R] and cannabinoid receptor type 2 [CB2R]. CB1R is most prominent in the central nervous system [CNS] and is associated with several psychiatric and neurological disorders.”

Line 76: Both anandamide and 2-AG are arachidonic acid derivatives.

 Answer to the comments on the introduction

We thank the referee for the precise comments that we have corrected in the introduction (See highlighted changes in the introduction to the manuscript, lines 66-76)

COMMENT: Line 86: Since dopamine plays a role in SUD, the synthesis of AEA due to dopaminergic receptor activation needs to be explained better, as it is an indirect mechanism. Furthermore, glutamate also plays a role in SUD, and endocannabinoids appear to modulate glutamatergic neurotransmission. Therefore, the authors should also consider adding some information about this.

Answer to the referee

As suggested we have incorporated a brief explanation on the engagement of the endogenous cannabinoid system to modulate the hyperactivity of both Dopamine and Glutamate transmission, that is enhanced in SUD and psychosis.

COMMENT: Line 92 to 94: I assume the authors intended to say something like: In general, the results of all these studies show an increased tone of the ECS in people with these mental disorders. This increased activation, especially in the early stages of the disease [REFs], may represent an attempt by the organism to counteract the neurophysiological changes triggered by the disease. The latter sentence is a strong statement. This has been shown for psychosis, e.g., “Koethe D, Giuffrida A, Schreiber D, et al. Anandamide elevation in cerebrospinal fluid in initial prodromal states of psychosis. The British Journal of Psychiatry 2009;194(4):371-2. doi:10.1192/bjp.bp.108.053843, but I am not sure if there is already enough data available supporting that this is also the case for depression and anxiety. Furthermore, references [20-25] refer to changes in depression and psychosis but not anxiety.  

Answer to the referee

We acknowledge the commentary of the referee. In fact the best association of mental disorder-plasma endocannabinoid concentrations,  is not with anxiety but with post-traumatic stress disorder. So we modified this paragraph, changing reference 21 and including reference Hill MN,  et al.  Reductions in circulating endocannabinoid levels in individuals with post-traumatic stress disorder following exposure to the World Trade Center attacks. Psychoneuroendocrinology. 2013 Dec;38(12):2952-61. doi: 10.1016/j.psyneuen.2013.08.004.

In addition, we have cited as new citation 25 the reference on the early changes in psychosis described by Koethe and colleagues and recommended by the referee.

COMMENTS TO Results:

The presentation of the sociodemographic data is confusing. At first, I thought there was no information on the type of SUD as this information is only included in table 3 but not in table 1. Combining tables 1 and 3 would facilitate understanding the cohorts' characteristics. Alternatively, information on psychiatric comorbidity, substance use, and psychotropic medication for the entire SUD cohort could be added to table 1.

Answer to the referee

We have added information about psychiatric comorbidity, substance use, and psychotropic medication for the entire SUD cohort in table 1

How long was the phase of abstinence? And how long did the included participants abuse substances before they were included in the study?

Answer to the referee

We have added both in Table 1 and Table 3 information on SUD duration years and days of abstinence for users, both those who were diagnosed with alcohol use disorder and those who were diagnosed with cocaine use disorder.

Lines 133-136: It would be easier to follow if you would immediately describe what type of changes have been observed instead of stating the results very generally and then adding a sentence like: “Thus, patients with SUD and antipsychotic treatment have more comorbidity substance use and psychiatric disorders.” This is only one example. I have recognized several times that it is possible to write a bit more to the point to facilitate the understanding of your highly interesting results.

Answer to the referee

At the beginning of this section we have included a paragraph in which, before summarizing the main significant differences and their implication, we describe how you suggest all the changes we have observed.

Line 151/152: What does “which confirmed some [of] the previous differences” mean? Can you please add a more detailed description of the results? I need to check the figure to understand that the AEA, OEA, and LEA levels are higher in people with SUD + antipsychotics than in people with SUD. Please also report the post hoc Sidak results.

Answer to the referee

Answer to the referee

In the new paragraph that we have included at the beginning of this section, it is detailed that the concentrations of these NAEs are higher in the SUD + antipsychotics group, so the last sentence can already be understood without having to check the figure. We have reported the post hoc Sidak results.

Line 177/178: How do the significant differences look like? Again, I need to check the figure. Please also report the post hoc Sidak results.

Answer to the referee

We have added a paragraph in this section where we describe the meaning of the differences found between the groups based on lifetime SUD and antipsychotic treatment. We have reported the post hoc Sidak results.

COMMENT TO Discussion:

Comparable to the introduction, the discussion is not easy to read, contains some typos, and requires careful revision.

For example, the first sentence should probably read: Two major findings emerge from the present study: first, confirmation of the value of monitoring NAEs in plasma in SUD as potential biomarkers, and second, the influence of psychiatric medications (in this case, atypical neuroleptics) on circulating blood levels of these lipid mediators.

 Answer to the referee

We have carefully edited the discussion for correcting these and other errors.

In the discussion and the abstract, it should be highlighted that the blood of abstinent individuals with CUD has been evaluated. The endocannabinoid system and, most likely, the lipidome as well are highly dynamic systems. Therefore, the disease phase will be important for interpreting the data.  

Answer to the referee

We have incorporated the recommended mention in both, abstract and discussion 

Line 245: I suggest adding the paper mentioned above here as well (Koethe et al. 2009). Furthermore, protective appears to be a better choice of words than defensive. Reference 39 reported that people with schizophrenia treated with atypical antipsychotics had CSF AEA levels comparable to untreated individuals and that both patient groups showed increased AEA levels compared to healthy volunteers. In contrast, the CSF AEA levels of individuals treated with typical antipsychotics did not differ from healthy volunteers. This is in contrast to the current findings and should be discussed.

 Answer to the referee

We have modified the discussion to consider the “protective” view of the endocannabinoid system. The reference 39 is discussed considering that

  1. a) Patients studied here are not primary psychosis but chronic SUD patients with psychotic symptomatology and chronically treated with neuroleptics,
  2. b) Differences may exist in between both studies, considering neuroleptic dose, length of treatment, dopamine D-2 preference binding in the typical versus multí-target preference on the atypical neuroleptics etc…

 Line 264/265: This is a very strong statement. Much more research is needed to be able to draw this conclusion. At least CSF AEA levels seem to be not affected by atypical antipsychotics.

Answer to the referee

We have modified this statement, considering the CSF findings.

Line 271: The disease state may play an important role. It also needs to be clarified if NAE changes are state or trait markers. Different baseline NAE levels may contribute to the development of SUD.

 Answer to the referee

We have considered the possibility of NAEs concentration being a trait marker. However, there is no agreement in preclinical studies if this is true or not. Both options might be possible, that elevated NAEs are a trait in addicts, and that the rise in plasma concentrations are directly derived of the pharmacological actions of acute/chronic drug exposure.

Line 276: To assess if NAEs are valuable biomarkers in SUD, studies are needed to evaluate NAEs in CSF, as it is still being determined if peripheral NAE levels also reflect changes in the brain.

 Answer to the referee

We introduced this recommendation as a limitation statement

Materials and Methods:

Statistics – It appears that the p-values have not been corrected for repeated testing, which should be done to compare plasma concentrations and the multiple ANCOVAs.

Answer to the referee

We have included in this section the formula for the corrected p-value according to the Sidak test.

Conclusion:

The conclusion is written very vaguely. I have no idea which changes were observed by reading the conclusion only.

We have rewritten the conclusion to indicate that the potential clinical utility of monitoring NAEs concentration is limited by the impact of certain drugs such as neuroleptics.

Reviewer 2 Report

The goal of this interesting study was to investigate the utility of plasma acethanolamides as biomarkers of substance use disorders as well as the current use of antipsychotic treatment as a potential confounding variable.

Basically, the methods are reasonable, the results of potential interest, and the rationale of this work is quite logical.

I have no major concerns to be addressed.

(Very) minor point: some style and spell check are required, like f.ex. page 9: "patients"  - line 244 "  and "antipsychotic" - line 264 to be corrected.

Author Response

REFEREE 2

The goal of this interesting study was to investigate the utility of plasma acethanolamides as biomarkers of substance use disorders as well as the current use of antipsychotic treatment as a potential confounding variable.

Basically, the methods are reasonable, the results of potential interest, and the rationale of this work is quite logical. I have no major concerns to be addressed.

(Very) minor point: some style and spell check are required, like f.ex. page 9: "patients"  - line 244 "  and "antipsychotic" - line 264 to be corrected.

Answer to Referee´s comments

We appreciate Referee’s comments. As suggested we did a thorough revision for spelling errors, as indicated
